# Phenotype-based cell-specific metabolic modeling reveals metabolic liabilities of cancer

Keren Yizhak[1]*[†], Edoardo Gaude[2†], Sylvia Le Dévédec[3], Yedael Y Waldman[1], Gideon Y Stein[4,5], Bob van de Water[3], Christian Frezza[2‡], Eytan Ruppin[1,5]*[‡]

[1]Blavatnik School of Computer Science, Tel-Aviv University, Tel-Aviv, Israel; [2]MRC Cancer Unit, University of Cambridge, Cambridge, United Kingdom; [3]Division of Toxicology, Leiden Academic Center for Drug Research, Leiden University, Leiden, Netherlands; [4]Department of Internal Medicine 'B', Beilinson Hospital, Rabin Medical Center, Petah-Tikva, Israel; [5]Sackler School of Medicine, Tel Aviv University, Tel-Aviv, Israel

**Abstract** Utilizing molecular data to derive functional physiological models tailored for specific cancer cells can facilitate the use of individually tailored therapies. To this end we present an approach termed PRIME for generating cell-specific genome-scale metabolic models (GSMMs) based on molecular and phenotypic data. We build >280 models of normal and cancer cell-lines that successfully predict metabolic phenotypes in an individual manner. We utilize this set of cell-specific models to predict drug targets that selectively inhibit cancerous but not normal cell proliferation. The top predicted target, *MLYCD*, is experimentally validated and the metabolic effects of MLYCD depletion investigated. Furthermore, we tested cell-specific predicted responses to the inhibition of metabolic enzymes, and successfully inferred the prognosis of cancer patients based on their PRIME-derived individual GSMMs. These results lay a computational basis and a counterpart experimental proof of concept for future personalized metabolic modeling applications, enhancing the search for novel selective anticancer therapies.

*For correspondence: kerenyiz@post.tau.ac.il (KY); ruppin@post.tau.ac.il (ER)

[†]These authors contributed equally to this work

[‡]These authors also contributed equally to this work

**Reviewing editor**: Chi Van Dang, University of Pennsylvania, United States

## Introduction

Personalized medicine is moving us closer to a more precise, predictable and powerful method of treatment, customized for the individual patient. One field of research in which personalized medicine holds great promise is cancer therapy. The use of molecular data to personalize cancer treatment and differentiate one type of cancer from another can facilitate the use of highly tailored therapies and offers tremendous potential for improved prognoses (*Simon and Roychowdhury, 2013*). A fundamental stepping-stone towards this goal is the ability to derive large-scale functional physiological models of specific cells that capture their unique cellular behavior. These models can then be utilized to identify drug targets that differentiate one cancer type from the other, and most importantly, distinguish them from their normal counterparts thus achieving treatment response selectivity.

This study addresses these challenges within the growing paradigm of Genome-Scale Metabolic Modeling, a computational framework for studying metabolism on a genome-scale that has been successfully used for a variety of applications (*Burgard et al., 2003*; *Oberhardt et al., 2009*; *Chandrasekaran and Price, 2010*; *Jensen and Papin, 2010*; *Lewis et al., 2010*; *Szappanos et al., 2011*; *Wessely et al., 2011*; *Agren et al., 2012*; *Lee et al., 2012*; *Lerman et al., 2012*; *Pey et al., 2012*; *Schuetz et al., 2012*; *Oberhardt et al., 2013*). In recent years, two Genome-Scale Metabolic Models (GSMMs) of human metabolism were published (*Duarte et al., 2007*; *Ma et al., 2007*), and

**eLife digest** Cancer is not just one disease, but a collection of disorders; as such there is no single general treatment that is effective against all cancers. Different tissues and organs—including the lungs, skin, and kidneys—can get cancer, and each need different treatments. Even two patients with the same type of cancer might respond differently to the same treatment.

Being able to distinguish between different cancer types would help doctors personalize a patient's cancer therapy—which would hopefully improve the outcome of the treatment. An important step in developing such personalized treatments is to find out how each type of cancer cell behaves and to see how this behavior differs both from normal, healthy cells and other types of cancer.

Countless chemical reactions take place inside living cells, and these reactions essentially dictate how a cell will grow and behave. The chemical reactions occurring inside a cancerous cell can be described as its 'metabolic phenotype' and will likely be different to the chemical reactions occurring in a healthy cell. Now Yizhak, Gaude et al. have used a range of data, including gene expression data, to create computer models of the metabolic phenotypes of 60 different types of human cancer cell. The same approach was also used to create metabolic models of over 200 healthy human cells that were dividing normally. Yizhak, Gaude et al. used these metabolic models to predict how quickly the different types of cancer cell would divide and how the cells would respond to drug treatments.

It may be possible to reduce the spread of all types of cancer—without also affecting healthy cells—by targeting proteins that help cancerous cells to proliferate. Yizhak, Gaude et al. used all of the models to search for genes that encode such proteins. One gene that was predicted to provide such a drug target encodes an enzyme that is needed to make and break down fatty acid molecules. Experiments confirmed that inhibiting this gene slowed the proliferation of both leukemia and kidney cancer cells, but had less of an effect on the growth of healthy bone marrow or kidney cells. Finally, Yizhak, Gaude et al. generated detailed metabolic profiles of cancer cells taken from over 700 breast and lung cancer patients and were able to use the models to successfully predict the outcome of the diseases in these patients.

Yizhak, Gaude et al.'s findings might help future efforts aimed at developing and delivering personalized cancer therapies. The next challenge is to use additional data—such as gene sequencing data—to generate more detailed and more accurate metabolic models for many cancer patients, to both predict their individual responses to available drugs and identify new patient-specific treatments.

their utility in predicting human metabolic phenotypes has been demonstrated in a wide range of studies (*Shlomi et al., 2008*; *Lewis et al., 2010*; *Folger et al., 2011*; *Frezza et al., 2011*; *Agren et al., 2012*; *Yizhak et al., 2013*). Recently, more comprehensive versions of the generic human model were published (*Thiele et al., 2013*; *Mardinoglu et al., 2014*). While these generic models are not specific to any cell- or tissue-type, they have successfully served both as a basis for generating context-specific models of tissues (*Shlomi et al., 2008*; *Jerby et al., 2010*; *Agren et al., 2012*) and for studying cancer metabolism (*Folger et al., 2011*; *Frezza et al., 2011*; *Shlomi et al., 2011*; *Agren et al., 2012*; *Facchetti et al., 2012*; *Wang et al., 2012*; *Dolfi et al., 2013*; *Agren et al., 2014*; *Yizhak et al., 2014*). Importantly, methods for building context-specific models do not take into account subtle differences in levels of expression of a particular enzyme, but rather its presence or absence. This coarse discretization makes these methods less applicable for the task of building cell-specific models, in cases where a high similarity in transcriptomics levels of different samples is observed. Namely, when the inter-individual variations in the molecular signatures of different cells are too small, this type of methods would lead to nearly identical models with little specific predictive value. Alternatively, absolute expression levels can be used to constrain the model's solution space, as previously done by E-Flux for studying bacterial metabolism (*Colijn et al., 2009*). Nonetheless, the applicability of E-Flux for studying human metabolism has not been established.

In this study we aim to derive cell-specific metabolic models for human cell lines that are capable of predicting metabolic phenotypes in an individual manner. We aimed to construct such models for the

human NCI-60 and HapMap cell line collections, where the similarity in expression levels of different cell lines is quite high. We began our investigation by testing the suitability of two existing model-building approaches towards this end. The moderate performance achieved by existing methods (see next section) have led us to develop a new cell-specific model building method termed PRIME (Personalized ReconstructIon of Metabolic models), which utilizes both molecular and phenotypic data for tailoring cell-specific GSMMs. We applied PRIME to reconstruct >280 GSMMs of cancer and normal proliferating cells, which are tested by their ability to predict metabolic phenotypes such as proliferation rate, drug response and biomarkers on an individual level. We then utilized the models of normal and cancer cell lines to predict cancer selective drug targets. We validate experimentally that the top predicted gene target, Malonyl-CoA decayboxylase (*MLYCD*), induces a clear selective effect on cell growth when tested in both leukemia and renal cancer cell lines, vs normal lymphoblast and renal cell lines. Furthermore, we used PRIME to reconstruct personalized metabolic models of breast and lung cancer patients successfully inferring their prognosis. We therefore suggest that PRIME can be applied in the future to a variety of personalized medicine applications where molecular and phenotypic data can be coupled together to find metabolic drug targets.

## Results

### Generation of a phenotype-based cell specific (PBCS) GSMMs via the PRIME approach

In this study we aim to derive individualized metabolic models for both normally proliferating lympho-blast cell lines (HapMap dataset), and a panel of cancer cell lines (the NCI-60 collection) (*Lee et al., 2007*; *Choy et al., 2008*). As these datasets contain both gene expression information and growth rate for each cell line, our goal has been to use the gene expression to build cell-specific models that can predict an array of metabolic phenotypes using the measured proliferation rates for initial testing and validation. The difference in the gene expression of HapMap and NCI-60 datasets is very subtle (mean Spearman R > 0.92, *Figure 1A*, upper panel), which may in turn imply that discretization-based methods would result here with nearly identical models that will fail to differentiate between their phenotypes. We therefore hypothesized that the integration of absolute expression levels would possibly be more suitable for our goal. To this end, we examined the performance of the two representative previously published methods on these datasets, one accepting discretized expression as inputs (iMAT [*Shlomi et al., 2008*]) and one analyzing the raw, non discretized expression data (E-Flux [*Colijn et al., 2009*]).

As shown in *Figures 1A and 2A*, The performance of these methods leaves much to be desired: iMAT, an omics-integration method that defines a subset of active and inactive reactions based on expression data (*Shlomi et al., 2008*), resulted in insignificant or even negative correlations between the actual and predicted proliferation rates for both datasets (HapMap: Spearman R = 0.03, p-value = 0.66; NCI-60: Spearman R = −0.07, p-value = 0.59, *Figure 1A* middle panel, *Figure 2A*), probably due to the high correlation in metabolic gene expression between samples (mean pair-wise Spearman R = 0.97 and R = 0.92 for the HapMap and NCI-60 datasets, respectively; *Figure 1A*). E-flux (*Colijn et al., 2009*) similarly failed to obtain significant results in predicting the HapMap cell lines' proliferation rates (Spearman R in the range of 0.1–0.11, p-value > 0.07, *Figure 1A* lower panel, *Figure 2A*, *Supplementary file 1A*), but obtained significant results in predicting the NCI-60 cell lines' proliferation rate (Spearman R in the range of 0.43–0.44, p-value > 3.6e-4, *Figure 1A* lower panel, *Figure 2A*, *Supplementary file 1A*).

We hence turned to develop a new approach termed PRIME that is designed for our specific task (*Figure 1B* and *Figure 1—figure supplement 1*). PRIME aims to reconstruct distinct, phenotype-based cell-specific metabolic models (PBCS) based on sample-specific molecular data. This is achieved by setting maximal flux capacity constraints *on a selected subset of reactions* in the generic species model, according to their associated gene expression levels and phenotypic data. PRIME's starting point is similar to E-Flux. While both methods utilize the rather straightforward notion of adjusting reactions' bounds according to expression levels, few key differences between them help PRIME generate more accurate models: (1) since modifying the reactions' bounds is considered to be a hard constraint, one should aim to avoid over-constraining the network based on irrelevant or noisy information. Clearly, only a subset of the metabolic genes affects a specific central cellular pheno-type. Accordingly, PRIME identifies this set in the wild type unperturbed case and modifies the bounds

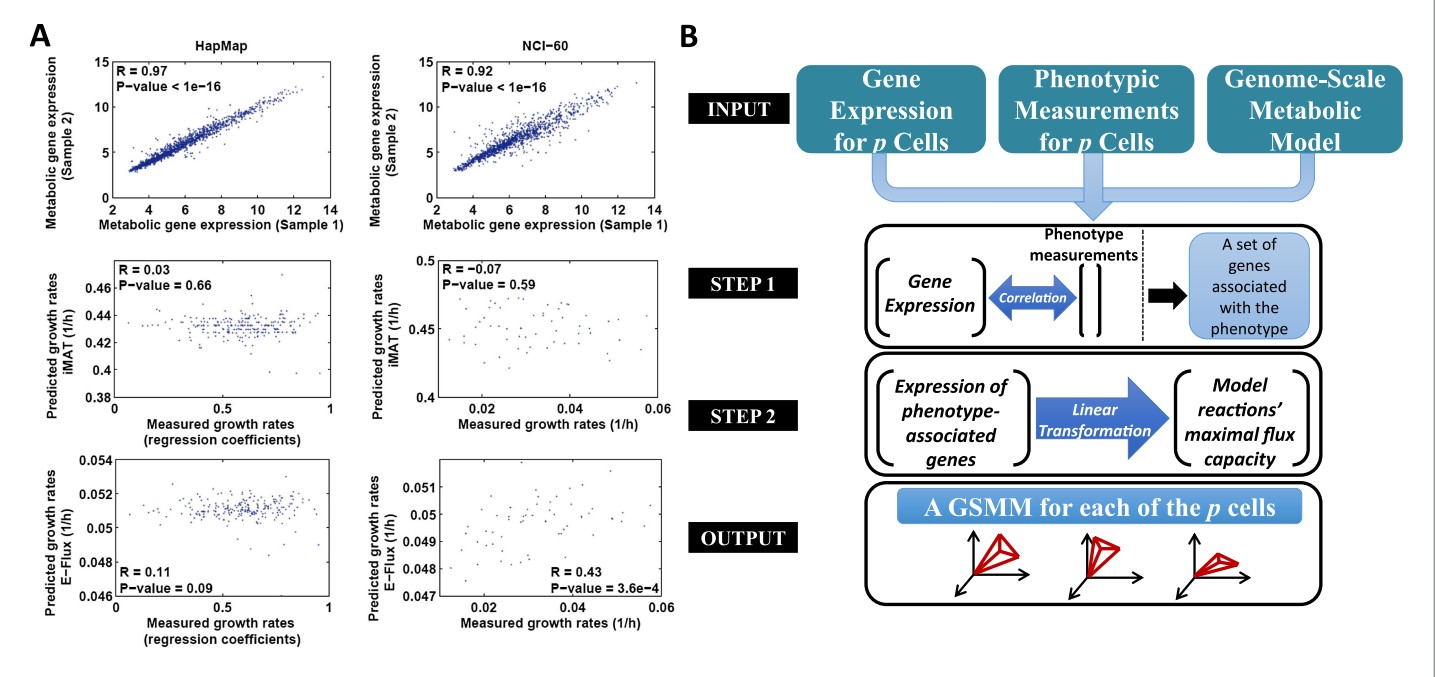

**Figure 1**. The PRIME pipeline and growth rate predictions obtained by different methods. (**A**) Upper panel: Spearman rank correlation between the metabolic gene expression of two representative cell lines in the HapMap (left) and NCI-60 (right) datatset (the two cell lines represent the average correlation across the entire datasets); Middle panel: Spearman rank correlation between predicted and measured growth rates in the HapMap (left) and NCI-60 (right) datatset as predicted by iMAT, a method that utilizes discrete gene expression signature as input; Lower Panel: Spearman rank correlation between predicted and measured growth rates in the HapMap (left) and NCI-60 (right) datatset as predicted by E-Flux, a method that utilizes absolute gene expression levels as input. (**B**) A schematic overview of PRIME. As input, PRIME gets a GSMM and gene expression measurements for p cells together with their associated phenotypic measurement (e.g., proliferation rate). (Step 1): A set of genes whose expression is significantly associated with the phenotype is identified. (Step 2): A linear transformation from the expression of the phenotype-associated genes, to reactions' upper bound (maximal flux capacity) is applied ('Materials and methods'). PRIME outputs a GSMM for each of the p input cells, such that each cell model generates a different feasible flux solution space. See also *Figure 1—figure supplement 1*.

The following figure supplement is available for figure 1:

**Figure supplement 1**. Biomass production as a function of flux upper bound.

of only the relevant set of reactions; (2) while a common assumption is that expression levels and flux rates are proportional, this is known to hold only partially (*Bordel et al., 2010*). PRIME therefore utilizes the additional phenotypic data to determine the direction (sign) of this relation and modifies the bounds accordingly ('Materials and methods'); (3) PRIME modifies reactions' bounds within a predefined range where the modification is known to have the greatest effect on a given phenotype ('Materials and methods'). Importantly, E-Flux has only been utilized to build models of two different bacterial conditions, by aggregating the expression levels of all samples associated with each condition. In this study we employ the principles described above to build individual cell models from the human metabolic model based on a *single sample* gene expression signature of each cell.

PRIME takes three key inputs: (a) gene expression levels of a set of samples; (b) a key phenotypic measurement (proliferation rate, in our case) that can be evaluated by a metabolic model; and (c) a generic GSMM (the human model, in our case). It then proceeds as follows: (1) A set of genes that are significantly correlated with the key phenotype of interest is determined (*Supplementary file 2A*); (2) The maximal flux capacity of reactions associated with the genes identified in (1) is modified according to the *directionality and level* of their corresponding gene expression level. Importantly, to assure that bound modifications would have an effect on the models' solution space, reactions' flux bounds are modified within an effective flux range. Accordingly, PRIME outputs a GSMM tailored uniquely for each input cell (see *Figure 1B*, *Figure 1—figure supplement 1* and the 'Materials and methods' for a formal description).

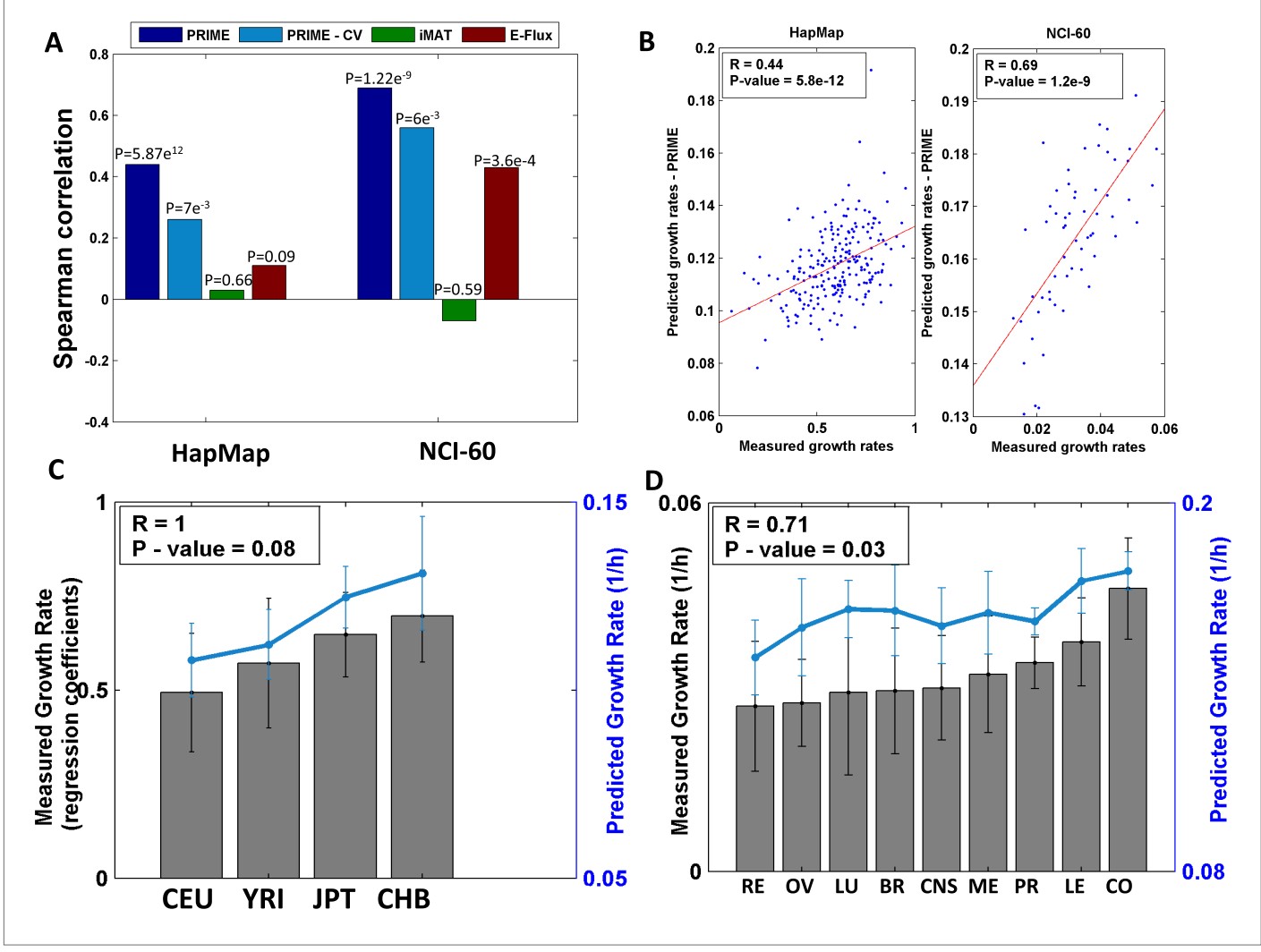

**Figure 2**. Growth rate predictions obtained by PRIME. (**A**) The Spearman correlation achieved by the different methods in predicting the individualized growth rates measurements across the HapMap and NCI-60 cell lines. (CV; Cross-Validation). (**B**) Individual predicted vs measured growth rates in the HapMap (left) and NCI-60 (right) datasets. (**C**) A comparison between mean predicted and measured growth rates across the four HapMap populations. Measured growth rates are represented as bars and the predicted growth rate is represented as a line. PRIME correctly predicts the population-based order of proliferation rates: CEU < YRI < JPT < CHB. (**D**) A comparison between mean predicted and measured growth rates across the nine tumor types composing the NCI-60 collection. Measured growth rates are represented as bars and the predicted growth rate is represented as a line (Spearman R = 0.71, p-value = 0.03); Leukemia (LE); Breast (BR); Central Nervous System (CNS); Colon (CO); Renal (RE); Lung (LU); Ovarian (OV); Prostate (PR); Melanoma (ME).

## PBCS metabolic models of normal lymphoblasts and cancer cell lines

We first applied PRIME to a dataset composed of 224 lymphoblast cell lines from the HapMap project (*International HapMap Consortium, 2005*). This dataset is composed of cell lines taken from healthy human individuals, from four different populations, including Caucasian (CEU), African (YRI), Chinese (CHB) and Japanese (JPT) ethnicities (*Supplementary file 1B*). Applying PRIME to the generic human model (*Duarte et al., 2007*), we constructed the corresponding 224 metabolic models, one for each cell line. The correlation between the proliferation rates predicted by these models and those measured experimentally is highly significant (Spearman R = 0.44, p-value = 5.87e-12, *Figure 2A–B*, *Supplementary file 1C* and *Supplementary file 2B*). In addition to capturing the differences between each of the cell lines the models also correctly predict the experimentally observed significant differences between populations' proliferation rates (CEU < YRI < JPT < CHB) in the correct order (*Figure 2C* and [*Stark et al., 2010*]). The correlation observed remains significant also after employing a five-fold

cross validation process 1000 times, controlling for the (indirect) use of proliferation rate in determining the modified reactions' set (mean Spearman R = 0.26, empiric p-value = 0.007, *Figure 2A*, 'Materials and methods'). Specifically, this analysis is performed by utilizing the set of growth-associated genes derived from the train-set to build the models of the test-set, where the correlation between measured and predicted proliferation rates is then evaluated.

We further applied PRIME to build individual models and predict the proliferation rates of 60 cancer cell lines, obtaining a highly significant correlation between the measured and predicted proliferation rates (Spearman R = 0.69, p-value = 1.22e-9, *Figure 2A–B*, *Supplementary file 1C* and *Supplementary file 2B*). A four-fold cross-validation analysis resulted with a mean Spearman correlation of 0.56 (empiric p-value = 0.006, *Figure 2A*, 'Materials and methods'). Grouping the samples into the nine tumor types found in this dataset and evaluating the mean proliferation rate of each group, a significant correlation is obtained between the measured and actual growth rates of the different tumors (Spearman R = 0.71, p-value = 0.03, *Figure 2D*). The higher correlation achieved for the cancer cell-lines in respect to that achieved for the normal cell-lines, is a result of the higher correlation found between metabolic gene expression and growth rate in the former datatset (see *Supplementary file 2A*).

To further examine the process employed by PRIME we tested three additional alternatives: (1) modifying the bounds of all enzyme-associated reactions and not only of those that are growth-related. This process decreased the correlation to Spearman R = 0.24, p-value = 2.4e-9 and Spearman R = 0.56, p-value = 2.8e-6 for the NCI-60 and HapMap datasets, respectively; (2) selecting random sets of reactions at the size of the original set and modifying their bounds according to their gene expression. Repeating this process 1000 times resulted with significantly inferior predictive performance in both datasets compared to PRIME (empiric p-value < 9.9e-4, 'Materials and methods'); (3) permuting the measured proliferation rates in each of the cell lines datasets for a 1000 times and correlating them with those computed by the PRIME models. In this case as well the original growth prediction results were found to be highly superior (empiric p-value < 9.9e-4, 'Materials and methods').

## Prediction of cell-specific metabolic liabilities using the NCI-60 collection

PRIME's major goal is to generate cell-specific metabolic models. Therefore, PRIME has the potential to guide pharmacological interventions based on the individual's phenotype, which underlies the basis of personalized medicine. We therefore tested the ability of PRIME to predict the response of each individual cell line to various metabolic drugs, and compared it with the response measured in vitro (*Scherf et al., 2000*; *Choy et al., 2008*; *Holbeck et al., 2010*; *Garnett et al., 2012*; *Lock et al., 2012*). *In silico* drug response is computed according to the biological phenotype measured experimentally, which in this case includes ATP levels, or AC50/IC50 values (the concentration at which a given compound exhibits half-maximal efficacy or half-maximal inhibition of cell growth, respectively). ATP flux production levels can be estimated directly in a metabolic model. The latter measurements (AC50/IC50) were computed by evaluating the flux through the drug's target reaction under 50% of drug maximal efficacy or 50% inhibition of cell maximal growth ('Materials and methods' and *Supplementary file 1D–F*). As shown in *Figure 3A*, this analysis yields a significant Spearman correlation (p-value < 0.05) between measured and predicted drug response for 12 out of 16 drugs tested in the HapMap and the NCI-60 datasets. Moreover, performing a permutation test in each of the datasets separately by permuting the measured drug response data, a highly significant result is obtained (empiric p-value < 9.9e-4, 'Materials and methods'). Applying a partial correlation analysis between *in silico* predicted and measured drug response while controlling for the experimentally measured proliferation rate (as growth rate itself has been implicated as a predictor of drug response, e.g., for cytotoxic drugs), we still find a significant association between predicted and measured drug response for the HapMap and CEU datasets, and in some cases even higher than before (*Supplementary file 1D–E*). These results demonstrate that utilizing a specifically-tailored metabolic model for predicting metabolic drugs response has a clear advantage over utilizing the raw data alone.

To further validate the NCI-60 PRIME models we have used measured uptake and secretion rates (*Jain et al., 2012*; *Dolfi et al., 2013*) and compared them to those predicted by our models ('Materials and methods'). We obtained significant Spearman correlations (Benjamini-Hochberg adjusted p-value with False Discovery Rate (FDR) and α = 0.05) for 14 out of 33 metabolites with a corresponding

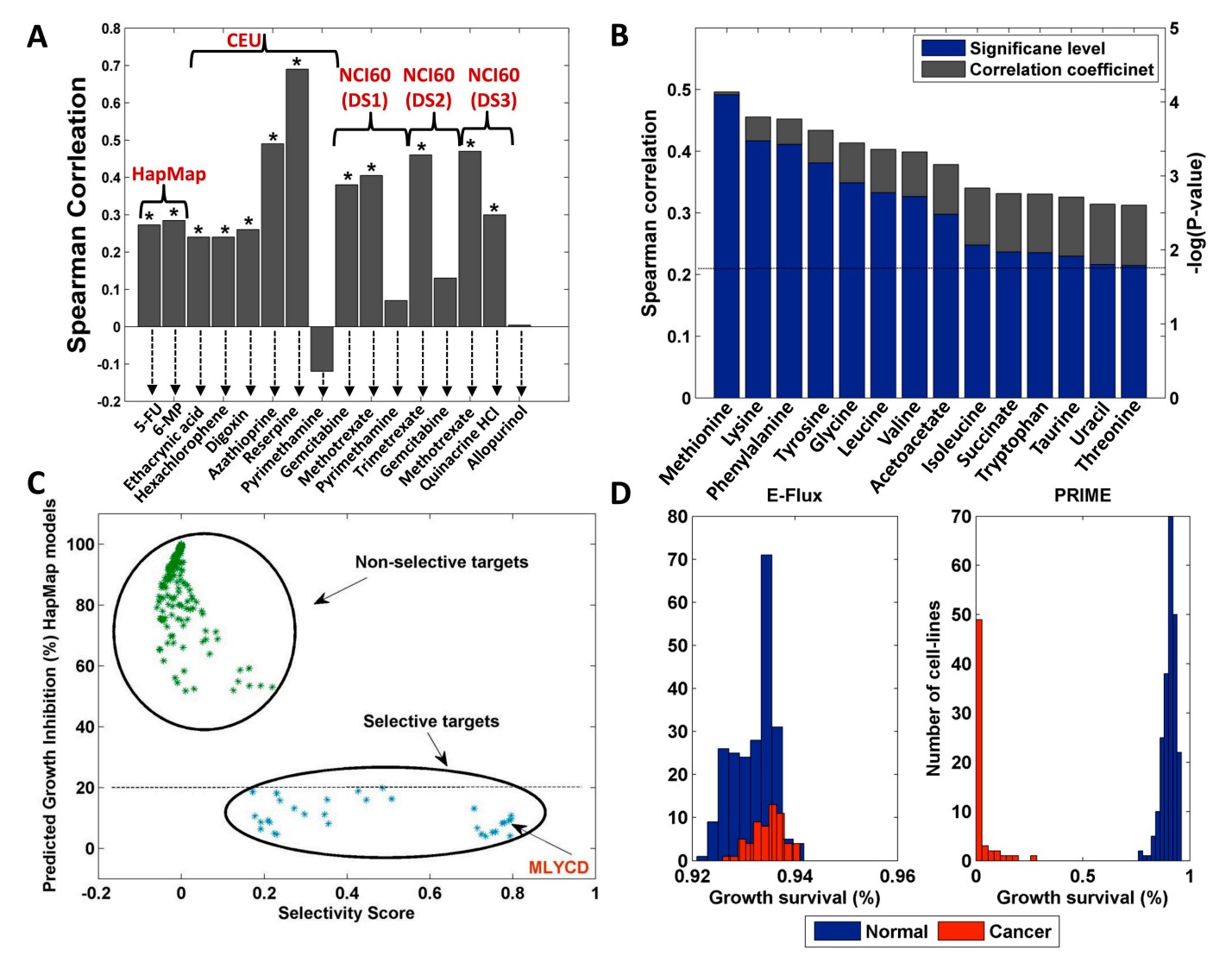

**Figure 3**. Drug response, biomarkers and selectivity analysis. (**A**) A comparison between measured and predicted drug response for the HapMap, CEU (Western European ancestry) and NCI-60 datasets. Overall, significant correlations (Spearman p-value < 0.05) were obtained for 12 out of the 16 drugs examined (those marked with an asterisk). The HapMap drugs are 5-fluorouracil (5FU) and 6-mercaptopurin (6MP); the CEU drugs are Ethacrynic acid, Hexachlorophene, Digoxin, Azathioprine, Reserpine and Pyrimethamine; The NCI-60 drugs for dataset 1 include Gemcitabine, Methotrexate and Pyrimethamine; For dataset 2, Trimetrexate and Gemcitabine; For dataset 3, Methotrexate, Quinacrine HCl and Allopurinol. (**B**) 14 metabolites for which a significant correlation between measured and predicted uptake and secretion rates is achieved. Both the Spearman correlation coefficient (gray) and the–log(p-value) (blue) are shown. The dashed line represents the FDR corrected significance level for α = 0.05. (**C**) Metabolic reaction targets that are predicted to be non-selective (green) or selective (blue). The x-axis represents the selectivity score ('Materials and methods') and the y-axis represents the growth inhibition predicted for the normal cell lines. Non-selective targets are predicted to reduce both normal and cancer cell growth by more than 50%. The selective targets are predicted to reduce normal cell growth by less than 20% and cancer cell growth by more than 30%. MLYCD is the third ranked target with a predicted reduction of >90% in cancer cell growth and <10% in normal cell growth. See also *Figure 3—figure supplement 1*. (**D**) Growth survival (in %) for the HapMap (normal) and NCI-60 (cancer) cell lines upon MLYCD knock down, as predicted by E-Flux and PRIME. While E-Flux predicts less than 10% reduction in cellular growth for both normal and cancer cell lines in a largely indiscriminate manner, PRIME predicts a cancer selective effect.

The following figure supplement is available for figure 3:

**Figure supplement 1**. Core metabolic pathways and their association with selective and non-selective predicted targets.

transporter reaction in the human model (*Figure 3B*). By performing a permutation test on the measured data a highly significant result is obtained (empiric p-value < 9.9e-4, 'Materials and methods'). Importantly, utilizing the models reconstructed by E-Flux for the same task, insignificant results are obtained for all metabolites.

The array of models built for both normal and cancer cells provides us with a unique opportunity not only to predict cell-specific drug target effects, but more importantly, to find drug targets that inhibit proliferation across all cancer cells but have no effect on the non-transformed counterpart. To this aim we simulated all knock downs of individual reactions in the 224 normal lymphoblasts and 60 cancer cell models, and quantified their selective effect on cell proliferation ('Materials and methods'). The set of predicted *non-selective* targets was highly enriched with current cytostatic drugs (*Wishart et al., 2008*; *Folger et al., 2011*) (mean hypergeometric p-value = 7.28e-4, *Figure 3—figure supplement 1* and *Supplementary file 1G*). Second, the predicted selective targets were enriched with targets of newly developed drugs (*Figure 3—figure supplement 1*): Out of the five metabolic enzyme drug targets reported in (*Cheong et al., 2012*), our analysis identified three as being *selective* (Hypergeometric p-value = 3.98e-4; *Supplementary file 2C*). To further validate these findings, we examined the clinical relevance of our predicted selective targets on a cohort of 1586 breast cancer patients (*Curtis et al., 2012*). A Cox multivariate regression analysis shows that this set is enriched (Hypergeometric p-value = 2.1e-5) with genes whose lower expression is significantly associated with improved survival (Benjamini-Hochberg adjusted p-values with FDR and α = 0.1, 'Materials and methods'), when examined together with known prognostic variables such as patients' clinical stage, histological grade, tumor size, lymph node status and estrogen receptor status. A similar analysis for the set of predicted *non-selective* targets yielded either borderline or insignificant results (*Supplementary file 1G*). A top predicted selective target is Malonyl-CoA Decarboxylase (*MLYCD*) (*Figure 3C*). While the highest ranked predicted reaction is catalyzed by isoenzymes and therefore more difficult to target experimentally, and the second ranked reaction occurs spontaneously, *MLYCD* is the first prediction that could be tested from a practical, experimental point of view (*Supplementary file 2C*). Of note, the knock down of MLYCD is predicted by E-Flux to reduce both normal and cancer cell proliferation by less than 10%, suggesting that without including phenotype-based constraints, this candidate gene would have not been revealed (*Figure 3D*). Interestingly, this enzyme has been recently proposed as potential anticancer target for breast cancer (*Zhou et al., 2009*), however its selective effects on other tumor types have not been assessed. Therefore, we decided to further investigate the role of MLYCD as selective target for cancer therapy.

## *MLYCD* selectively suppresses cancer cell proliferation

The prediction of selective targets made by PRIME capitalizes on the non-transformed lymphoblast cell lines HapMap as *normal* counterpart. Therefore, to experimentally validate the *cancer versus normal* selectivity, we initially used leukemia cells, the only hematological tumor type in the NCI-60 database. In line with PRIME's predictions, the small interfering RNA (siRNA)-mediated silencing of *MLYCD* significantly inhibited the proliferation of the leukemia cell lines RPMI-8226 and K562 cells, but had no effect on HapMap cells (*Figure 4A–B*). To further corroborate the *cancer versus normal* selectivity, we tested the effects of *MLYCD* depletion on two renal cancer cell lines, TK-10 and CAKI-1, using the non-transformed renal cell line HK-2 as *normal* control (*Figure 4C*). Of note, the silencing of *MLYCD* suppressed proliferation of renal cancer cell lines without affecting the non-transformed counterpart (*Figure 4D*). Importantly, the anti-proliferative effects of MLYCD suppression could not be explained by the different expression of the enzyme among the different cell lines (*Figure 4—figure supplement 1*). These results substantiated PRIME's prediction that MLYCD is a cancer selective drug target.

## Silencing of MLYCD deregulates fatty oxidation and TCA cycle

We wanted to functionally validate the effect of silencing of MLYCD in cancer cells. To this aim, we generated a leukemia cell line that stably expresses a doxycycline-inducible short hairpin RNA (shRNA) targeting *MLYCD*. The incubation with doxycycline resulted in efficient silencing of *MLYCD* and led to a significant growth inhibition (*Figure 5—figure supplement 1–2*), in line with the siRNA experiments. Previous reports have shown that MLYCD depletion leads to the accumulation of malonyl-CoA, which blocks fatty acid oxidation by allosteric inhibition of the mitochondrial enzyme Carnitine-Palmitoyl-Transferase (CPT1) (*Zhou et al., 2009*). These observations prompted us to investigate the effects of

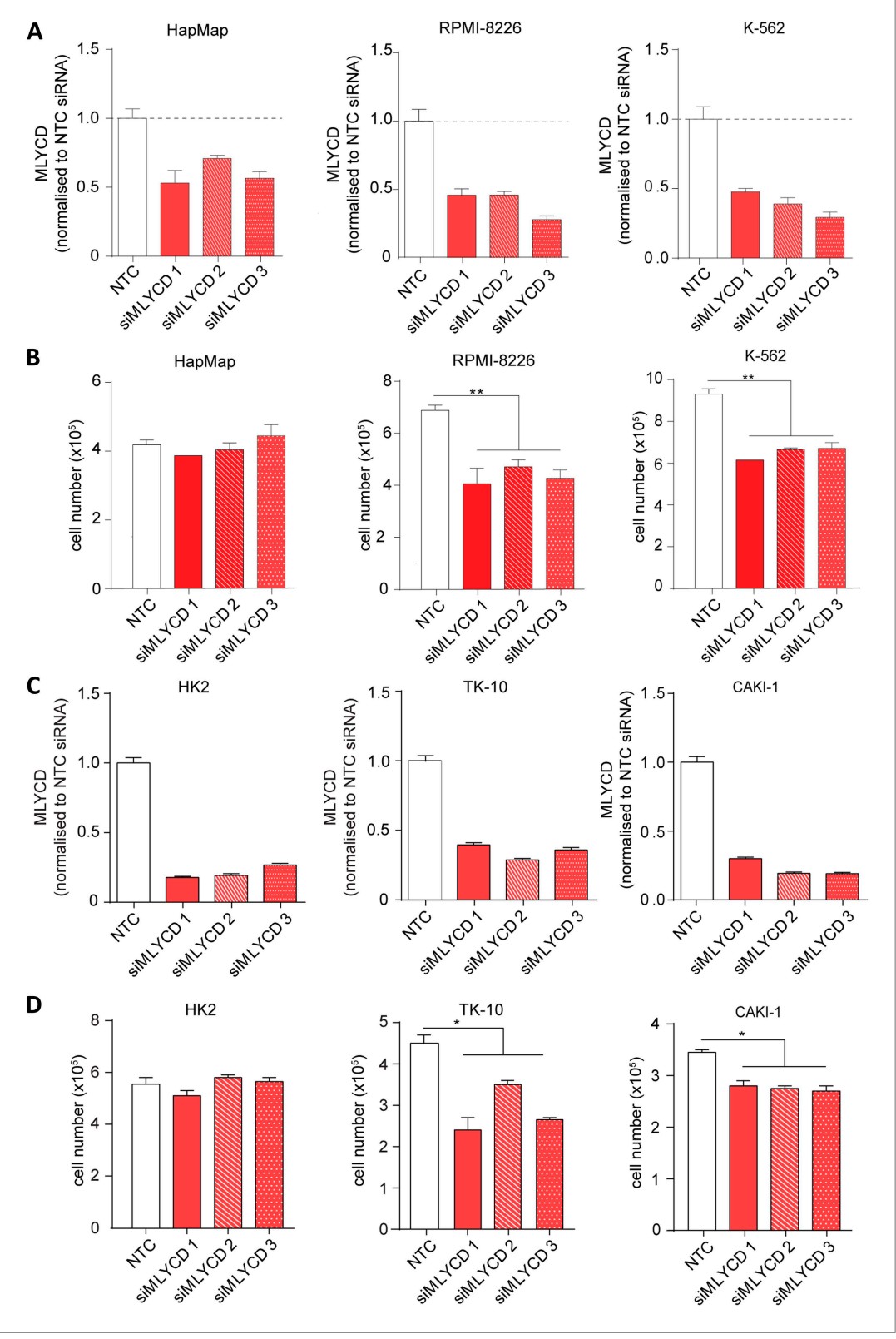

**Figure 4**. *MLYCD* depletion on normal and cancer cell lines. (**A**) *MLYCD* mRNA expression upon nucleofection with Non Targeting Control (NTC) and three independent siRNA constructs in HapMap, RPMI-8226 and K562 cells. (**B**) Cell counts after 72 hr of culture of the indicated cell lines. (**C**) *MLYCD* mRNA expression upon nucleofection

*Figure 4. Continued on next page*

*Figure 4. Continued*

with Non Targeting Control (NTC) and three independent siRNA constructs in HK2, TK10 and CAKI1 cells. (**D**) Cell counts after 72 hr of culture of the indicated cell lines. Data are shown as mean ± s.e.m of three independent cultures. *p-value<0.05. **p-value<0.01. ***p-value < 0.001.

The following figure supplement is available for figure 4:

**Figure supplement 1**. Expression levels of *MLYCD* across multiple cancer and normal cell lines.

the loss of MLYCD on fatty acid oxidation. To this aim, cells were incubated with $^{13}C_{16}$-palmitate and the abundance of $^{13}C$-labelled palmitoyl-carnitine and of TCA cycle metabolites was measured by liquid chromatography coupled to mass spectrometry (LCMS) (see *Figure 5A* for a schematic of the experiment). We observed a significant decrease in the $^{13}C$-labelling of palmitoyl-carnitine (*Figure 5B*) and of the m+2 isotopologues of TCA cycle intermediates (*Figure 5C*, and *Figure 5—figure supplement 3* for the full isotopologue analyses of these metabolites), indicating that fatty acid oxidation is reduced in MLYCD-depleted cells. Of note, this marked decrease in fatty acid oxidation only partially affected the overall abundance of TCA cycle intermediates (*Figure 5—figure supplement 4*). We also noticed a striking accumulation of succinate and a decrease in fumarate and malate in MLYCD-depleted cells (*Figure 5—figure supplement 4)*. These results are consistent with the inhibition of the TCA cycle enzyme succinate dehydrogenase (SDH), which may be caused by malonyl-CoA-derived malonate. Taken together, these results show that the silencing of *MLYCD* is sufficient to inhibit fatty acid oxidation and alter TCA cycle.

## Silencing of MLYCD accelerates fatty acid synthesis and increases the demands of reducing power

We then used the PRIME-derived models to systematically assess the metabolic changes that occur upon MLYCD inactivation. Of note, the model predicted that upon *MLYCD* suppression, part of the accumulated malonyl-CoA is diverted to fatty acid biosynthesis. Since this process requires NADPH as source of reducing power, the aberrant activation of fatty acid synthesis caused by the loss of MLYCD would impair redox homeostasis of the cell (*Berg, 2002*) (*Figure 5—figure supplement 5* and *Supplementary file 1H*). We validated this hypothesis by first assessing fatty acid synthesis. To this aim, cells were incubated with $^{13}C_6$-glucose and the abundance of $^{13}C$-labelled TCA cycle intermediates and palmitate were analyzed by LCMS (*Figure 5D*). While the labeling of citrate, the main lipogenic precursor, was, if any, slightly decreased (*Figure 5E*), the m+4 and m+6 isotopologues of palmitate were significantly increased in MLYCD-depleted cells (*Figure 5F*), suggesting that fatty acid synthesis is accelerated in these cells. Of note, the reduction of the m+2 and m+4 isotopologues of TCA cycle intermediates suggested that the oxidative capacity of the TCA cycle is intact, albeit reduced, in MLYCD-depleted cells (*Figure 5—figure supplement 6*). To validate the prediction that MLYCD-depleted cells increase the demand of NAPDH to fuel fatty acid synthesis, we measured the activity of the pentose phosphate pathway (PPP), the major source of cytosolic NADPH (*Fan et al., 2014*). To this end, cells were incubated with 1,2-$^{13}C_2$-glucose and the amount of singly (m+1) or doubly (m+2) labeled lactate was used as measure of PPP or glycolysis activity, respectively (see *Figure 5G* for a representation of the experiment). As predicted by PRIME, PPP flux was increased in MLYCD-depleted cells (*Figure 5H–I*). Together, these results corroborate the prediction made by PRIME that the loss of MLYCD increases fatty acid synthesis and impinges on the PPP for generation of reducing power. Finally, we tested whether the observed activation of fatty acid synthesis, by draining NADPH, impairs the capacity of cells to maintain redox homeostasis. In line with this hypothesis, MLYCD-depleted cells exhibited a lower GSH/GSSG ratio compared to control cells (*Figure 5I*). Furthermore, the incubation of cells with the antioxidant N-acetyl-cysteine (NAC) fully restored the proliferation defects observed in MLYCD-depleted cells (*Figure 5J*). Taken together, these results suggest that the suppression of cancer cell proliferation caused by the loss of MLYCD depends, at least in part, on the aberrant activation of fatty acid synthesis, which leads to a reduced ability of cells to maintain redox homeostasis. Overall, this investigation showed the benefits of PRIME to predict and investigate metabolic liabilities of cancer cells, based on cell-specific metabolic models.

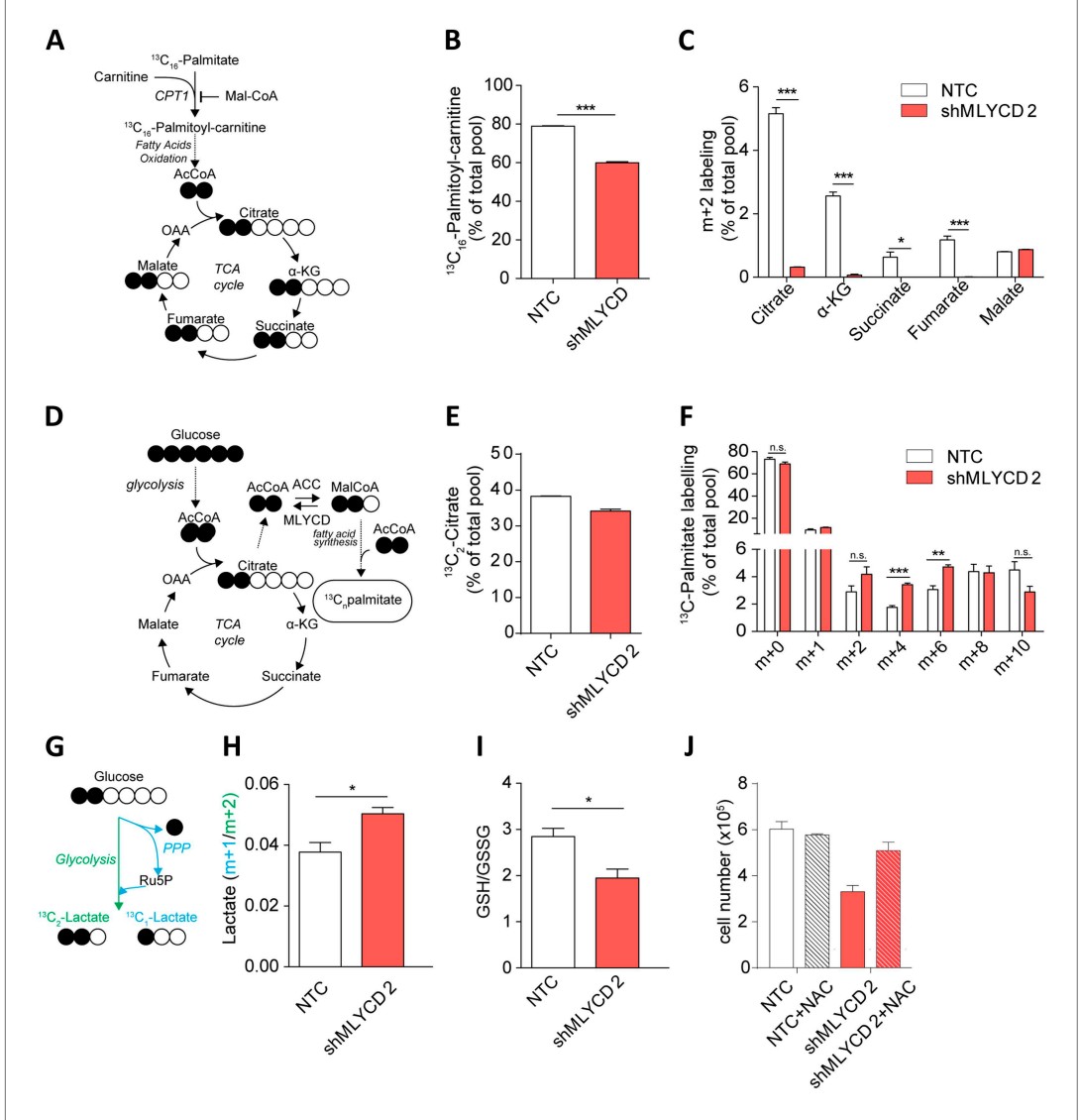

**Figure 5**. Metabolic characterization of *MLYCD* depletion. (**A**) Schematic representation of isotope tracing experiment with $^{13}C_{16}$-Palmitate. Black-filled circles indicate $^{13}C$-carbon, whereas the white filled circles represent the unlabeled carbon. The schematic shows the expected composition of labeled carbons of the indicated metabolites. (**B**) Labeling incorporation from $^{13}C$-Palmitate into Palmitoyl-carnitine in non-targeting control (NTC) and MLYCD-depleted (shMLYCD) cells. Data are shown as percentage of $^{13}C_{16}$-palmitoylcarnitine to the total pool of Palmitoyl-carnitine. (**C**) Labeling incorporation from $^{13}C_{16}$-palmitate into TCA cycle intermediates of the indicated cell lines. Data are shown as percentage of the m+2 isotopologue to the total pool size of each metabolite. (**D**) Schematic representation of isotope tracing experiment with $^{13}C_6$-Glucose. The distribution of light and heavy carbons is depicted as in **A**. (**E**) Labeling of Citrate and of (**F**) Palmitate after incubation with $^{13}C_6$-glucose. Data are shown as percentage of the indicated isotopologue to the total pool size of each metabolite. Isotopologue distribution of citrate is indicated in *Figure 5—figure supplement 6*. Palmitate isotopologues above m+10 were not detected (**G**) Schematic representation of isotope tracing experiment with 1,2-$^{13}C_2$-Glucose. Ru5p: ribulose-5-phosphate. The distribution of light and heavy carbons is depicted as in **A**. (**H**) Ratio between m+1 and m+2 isotopologues of Lactate in the indicated cell lines. (**I**) Ratio between reduced (GSH) and oxidized (GSSG) glutathione in RPMI-8226 cells infected with the indicated constructs. (**J**) Cell counts after 72 hr of culture of the indicated cell lines in the presence or absence of 2 mM N-Acetyl Cysteine. Data are shown as mean ± s.e.m of three independent cultures. *p-value<0.05. **p-value<0.01. ***p-value < 0.001.

*Figure 5. Continued on next page*

*Figure 5. Continued*

The following figure supplements are available for figure 5:

**Figure supplement 1**. Silencing of *MLYCD* in RPMI-8226 cells using shRNA.

**Figure supplement 2**. Effects of Silencing of MLYCD in RPMI-8226 cells.

**Figure supplement 3**. Isotopologue distribution of TCA cycle intermediates after incubation with $^{13}C_{16}$-palmitate.

**Figure supplement 4**. LCMS analyses of TCA cycle intermediates in MLYCD-depleted cells.

**Figure supplement 5**. A schematic description of the metabolic changes following MLYCD knock down.

**Figure supplement 6**. TCA cycle activity in MLYCD-depleted cells.

## Predicting gene knock downs that differentially modulate breast cancer cells growth

We next aimed to go beyond predicting targets that are selective with respect to *cancer versus normal* cell populations as a whole, to study if we can use PRIME to predict the differential response amongst cancer cell lines to specific treatments. To this end we used PRIME models of individual breast cancer cell lines of the NCI-60 panel, and simulated via Minimization of Metabolic Adjustment (MOMA) (*Segre et al., 2002*) the knock down of all metabolic reactions catalyzed by a single gene, examining their effect on cell growth ('Materials and methods'). We focused on reactions whose knock down yielded highly variable predicted growth rates across the different cell lines studied. 13 genes associated with these top ranked reactions and spanning different metabolic pathways were selected for further experimental investigation ('Materials and methods' and *Supplementary file 2D*). The effect of each of these genes on cell growth was examined via small interference RNA (siRNA) knock down in the two cell lines predicted to have the most differential effect on cell growth. 11 out of the 13 genes studied were found to have an effect on cell growth as predicted by the models (*Figure 6A* and *Supplementary file 2D*, empiric p-value < 0.01, 'Materials and methods'). A significant correlation is obtained between predicted and measured % inhibition values across all 11 targets (Spearman R = 0.64, p-value = 1e-3). These data underscore the ability of PRIME to successfully predict individual cell-specific responses of cancer cells to the knock down of metabolic enzymes, at least at a qualitative level.

## Reconstructing personalized metabolic models of breast and lung cancer patients

Finally, we examined PRIME's ability to build personalized models of cancer patients and predict their prognosis based on gene expression levels collected from biopsy samples. Importantly, growth rate measurements are not available for these datasets. Nonetheless, a possible way to overcome this hurdle and to build personalized metabolic models for cancer patients is to use phenotypic data measured for one set of cells to reconstruct models of a different set of cells or clinical samples. To examine this approach we utilized the set of growth-associated genes derived from the NCI-60 collection to build personalized GSMMs of more than 700 breast and lung cancer clinical samples (*Miller et al., 2005*; *Chang et al., 2010*; *Okayama et al., 2012*). A Kaplan–Meier survival analysis (*Kaplan and Meier, 1958*) showed that patients with predicted low growth rate had significantly improved survival compared to those with a predicted high growth rate (logrank p-values are: 0.01, 1e-3 and 0.02 for Miller et al., Chang et al. and Okayama et al. respectively, *Figure 6B*, *Supplementary file 1I*, 'Materials and methods'). This result was further supported by a Cox univariate survival analysis (*Grambsch, 2000*) (p-values are: 1e-3, 1e-4 and 2e-3 for Miller et al., Chang et al. and Okayama et al. respectively, *Supplementary file 1I*) and by performing a permutation test (p-values are: 0.015, 2e-3 and 0.018 for Miller et al., Chang et al. and Okayama et al. respectively, 'Materials and methods'). Of note, estimating the samples growth rates directly from the gene expression data by using multiple linear regression, resulted in inferior performance (*Supplementary file 1J*), testifying to the added value of personalized GSMMs. Importantly, while iMAT and E-Flux require only 'omics' data and can hence

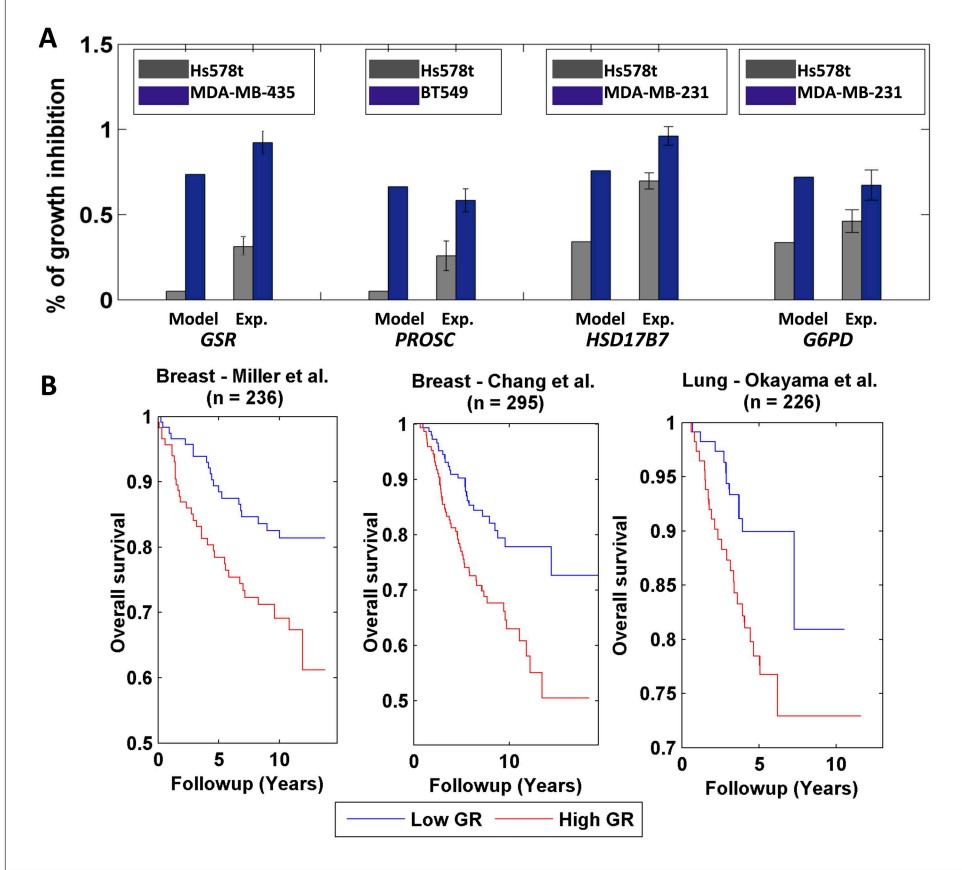

**Figure 6**. Differential growth affects in breast cancer cell-lines and clinical data analysis. (**A**) Four gene/reaction targets showing a differential effect on cancer cell growth (represented as % of growth inhibition) according to both PRIME's predictions and experimental validations via siRNA knock downs (when compared to a negative control, a siRNA that targets luciferase). Each gene was tested experimentally in two cell lines in triplicate, where the gene knock down is predicted to have the lowest and highest effect on cell growth. 11 out of the 13 top predictions tested were confirmed experimentally. Data are shown as mean ± s.e.m. For the full list see *Supplementary file 2D*. The genes GSR and PROSC are predicted to completely suppress the Hs578 t cell line growth (*Supplementary file 2*) but for presentation appear with a 0.05% height bar; (**B**) Kaplan-Meier plots for the two breast cancer datasets and for a lung cancer dataset. In all cases low growth rate (GR) is associated with improved survival.

be applied directly, they fail to obtain meaningful and significant results in this setting as well (*Supplementary file 1K*).

## Discussion

In this study we present a novel method termed PRIME for building cell-specific GSMMs based on the integration of gene expression and phenotypic data. We apply this method for the reconstruction of metabolic models of both cancer and normal cells. To the best of our knowledge, PRIME is the first method able to generate human cell-specific GSMMs that can predict metabolic phenotypes in an individual manner, including growth rates and drug response. The set of normal and cancer PRIME-derived models is utilized to identify a set of drug targets that can inhibit the proliferation of specific cell lines, as well as metabolic targets that can selectively block cancer but not normal cells growth. The experimental validation that we provide testifies that coupling molecular and phenotypic data for building cell-specific models can enhance the predictive power of GSMMs.

As many other computational approaches, PRIME is not devoid of limitations. First, PRIME assumes that cells try to maximize their proliferation, while different objective function(s) should be considered for non-proliferating cells. Second, we assume that all models share the same set of enzymes and differ only in their cellular abundance, but different cells may express different coding variants that should

be incorporated in future studies. Third, PRIME relies on the measurement of a specific phenotype that is not always available for a given set of cells or samples. Here we introduced a possible way to overcome this hurdle, as demonstrated by PRIME's ability to utilize clinical data and build cell-specific GSMMs tailored for each individual patient. However, while this analysis provided significant results, the obtained signal is mild and the question whether and how best one can identify a universal set of growth-associated genes still requires further study. Given the results obtained, one can confidently expect that follow-up work analyzing richer datasets, and most importantly, incorporating additional kinds of omics data (such as enzyme sequence data) will significantly improve the predictive power of PRIME further.

In this work we have also experimentally validated the prediction made by PRIME that MLYCD inhibition selectively affects cancer proliferation. MLYCD is an important enzyme of fatty acid metabolism, which role in cancer therapy has been recently suggested (*Zhou et al., 2009*). However, the selectivity across cancer types, and the mechanism of action of its inhibition have not been fully investigated. Our results show that the silencing of *MLYCD* has an anti-proliferative effect across multiple cancer cell lines but spares the non-transformed counterparts, confirming PRIME's predictions. We have also shed some light on the functional effects of inactivation of MLYCD in cancer cells. The toxic effects of MLYCD inhibition have been previously attributed to the accumulation of malonyl-coA and to the inhibition of fatty acid oxidation (*Zhou et al., 2009*). Our results suggest that, besides turning off fatty acid oxidation and partially deregulating TCA cycle, the loss of MLYCD stimulates fatty acid synthesis, which drains reducing equivalents and sensitize cells to oxidative stress. Therefore, our results not only confirmed the *cancer versus normal* selectivity of MLYCD inhibition but also elucidated a novel liability of cancer cells based on the pharmacological inhibition of fatty acid metabolism. Of note, both these features were accurately predicted by PRIME. Importantly, in humans, the loss of MLYCD leads to methylmalonic aciduria, an extremely rare autosomal recessive disorder. Nevertheless, in vivo experiments in rodents and pigs (*Dyck et al., 2004*; *Wu et al., 2014*), ex vivo experiments in human skeletal muscle (*Bouzakri et al., 2008*), and in MRC-5 non-transformed fibroblasts (*Zhou et al., 2009*) suggest that the inhibition of MLYCD is well tolerated, as our results indicate. It is therefore possible that the inhibition of the enzyme has no detrimental effects on normal cells and tissues, and that other factors contribute to the severity of MLYCD deficiency in humans, including a toxic effect of the mutated protein (*Polinati et al., 2014*).

In summary, we here show that incorporating gene expression measurements and phenotypic data within a genome-scale model of human metabolism via PRIME results in functional cell-specific models with considerable predictive power. We believe that the demonstrated ability of PRIME to predict the effects of known metabolically-targeted drugs on individual cell proliferation rates will help to pave the way for tailoring specific therapies based on metabolic modeling of cancer biopsies from individual patients.

## Materials and methods

### A constraint-based model (CBM) of metabolism

A metabolic network consisting of $m$ metabolites and $n$ reactions can be represented by a *stoichiometric matrix S*, where the entry $S_{ij}$ represents the stoichiometric coefficient of metabolite $i$ in reaction $j$ (*Price et al., 2004*). CBM imposes mass balance, directionality and flux capacity constraints on the space of possible fluxes in the metabolic network's reactions through the set of linear equations:

$$S \cdot v = 0 \tag{1}$$

$$v_{min} \leq v \leq v_{max} \tag{2}$$

where $v$ is the flux vector for all of the reactions in the model (i.e., the *flux distribution*). The exchange of metabolites with the environment is represented as a set of *exchange (transport) reactions*, enabling a pre-defined set of metabolites to be either taken up or secreted from the growth media. The steady-state assumption represented in *Equation (1)* constrains the production rate of each metabolite to be equal to its consumption rate. Enzymatic directionality and flux capacity constraints define lower and upper bounds on the fluxes and are embedded in *Equation (2)*. The biomass function utilized here is taken from (*Folger et al., 2011*). The media simulated in all the analyses throughout the paper is the RPMI-1640 media that was used to grow the cell lines experimentally (*Lee et al., 2007*; *Choy et al., 2008*).

Gene knock outs are simulated by constraining the flux through the corresponding metabolic reaction to zero. Following, two different approaches can be taken to estimate the effect of a perturbation on the network: (1) via Flux Balance Analysis (FBA) (*Varma and Palsson, 1994*) where maximization of growth rate is defined as the cellular objective function (max $V_{bio}$); (2) Minimization of Metabolic Adjustment (MOMA) (*Segre et al., 2002*) where the minimization of the Euclidean distance between a wild-type flux distribution ($V_{wt}$) and the post-perturbation flux distribution ($V_{KO}$) is set as the cellular objective function $min\sqrt{\sum_{i}^{n}\left(V_{wt,i}-V_{KO,i}\right)^2}$. Different wild-type flux distributions are obtained via sampling where each sample is determined based on a FBA analysis maximizing for cellular growth.

## The PRIME algorithm

PRIME is given the following three inputs: (1) a set of *p* samples with gene expression levels; (2) the *p* samples' corresponding growth rate measurements; and (3) a generic model (the human model, in our case). Next, the model reconstruction process is as follows:

1. Each reversible reaction is decomposed into its forward and backward direction and the maximal biomass production is evaluated. Next, the upper bound of all the reactions in the network is decreased simultaneously in steps of 0.1. In each step, the maximal biomass production is re-evaluated and the process proceeds as long as the reduction in bound doesn't decrease the maximal production found above by more than an ε (here we used ε = 1e-4). Finally, the upper bound of all reactions is set to the minimal upper bound allowed by this process. The goal of this step is only to narrow down the solution space and reduce the effect of futile cycles in the simulation of gene perturbation.
2. Next, the correlation between the expression of each reaction in the network and the measured growth rates is evaluated. The expression of a given reaction is defined as the mean expression of its catalyzing enzymes. The significance threshold is corrected by FDR with α = 0.05.
3. The upper bound of each reaction demonstrating a significant correlation to the growth rate (e.g., *t* reactions) is modified in a manner that is linearly related to its expression value. Specifically, we generate the Exp-matrix (*E*), a (*t* × *p*) matrix that embeds the information on the direction and magnitude of change of the upper bound based on the expression data. For each reaction *a* in sample *b* we define the Exp-matrix such that:

$$E_{a,b} = \frac{\rho_a}{|\rho_a|} \cdot GE_{a,b} \tag{3}$$

In *Equation (3)*, $GE_{a,b}$ represents the expression value of reaction *a* in sample *b*. Likewise, $\rho(a)$ represents the correlation coefficient of reaction *a* as found in step (2). Overall, for reactions whose expression is positively correlated with growth rate, the corresponding values in the matrix increase (become more positive) as the expression increases. Alternatively, for negatively correlated reactions, the corresponding values in the matrix decrease (become more negative) as the expression increases (due to the multiplication by $\frac{\rho_a}{|\rho_a|}$ which equals to −1 in this scenario). We then apply *Equation (4)* to normalize the values of the Exp-matrix and adapt them to the actual upper bounds. In this normalization procedure each reaction *a* is normalized across its *p* samples such that the bound associated with the sample having the lowest (highest) expression value is assigned the minimal (maximal) value of the normalization range, respectively.

$$UB_{a,b} = \left(\frac{E_{a,b} - min(E_a)}{max(E_a) - min(E_a)} \cdot \left(maxNormVal - minNormVal\right)\right) + minNormVal \tag{4}$$

$min(E_a)$ and $max(E_a)$ refer to the minimal and maximal value of reaction *a* across all *p* samples in the Exp-matrix, respectively. The minimal and maximal values of the normalization range (*minNormVal* and *maxNormVal*, respectively) are determined according to the procedure described in the next section.

## Defining the PRIME normalization range

1. *minNormVal* is set to be the minimal flux necessary for biomass production. This value is computed in the following manner: First, the set of essential reactions in the model is identified via Flux Balance

Analysis. This set is composed of those reactions that their knock out reduces growth by more than 90% of its maximal rate. Next, the minimal flux through each essential reaction is found via Flux Variability Analysis (*Varma and Palsson, 1994*). As each of these reactions is necessary for biomass production, reducing the upper bound below their minimal flux value would result with a lethal phenotype. We therefore set *minNormVal* to be the maximal value among these values (*Figure 1—figure supplement 1*).

2. To define the maximal value of the normalization range (*maxNormVal*) we examine the change in biomass production as a function of the model's upper bounds according to the following steps:

   A. First, we define the set of reactions in the model that are significantly correlated to the proliferation rate (as described in step (2) of PRIME above).
   B. Next, we examine how the biomass production is changed as a function of the model's upper bound. This is done by changing the upper bounds of the growth-associated reactions in steps of 0.1, and in each step re-evaluating the biomass production.
   C. Lastly, *maxNormVal* is defined as the maximal value beyond which the change in biomass production decreases (*Figure 1—figure supplement 1*).

Importantly, applying alternative ranges resulted with less optimal results in all datasets analyzed here.

The PRIME code and the generated models are provided as *Supplementary file 3 and 4*, respectively.

## Cross validation and permutation test

K-fold cross validation analysis is done by splitting the samples of the examined dataset to train- and test-sets. The set of growth-associated reactions found in the train-set is then used to build the models of the test-set. The correlation reported is the mean Spearman correlation achieved by comparing the measured and predicted growth rates of the test-set alone, while repeating this process 1000 times. The empiric p-value is computed by permuting the gene expression 1000 times, in each case building the resulting models and performing the cross-validation analysis as described here. Finally we compared the resulting mean Spearman correlation of each of these models to that obtained with the original data. Generally, all permutation tests are repeated 1000 times. Empiric p-value is then computed as $(n+1)/1001$ where n equals the number of times a random set of values yields a result which is more significant than the original result obtained when the data is not permuted.

## Drug response simulations

Each drug is mapped to its corresponding metabolic reaction through its known enzymatic targets according to DrugBank database (*Wishart et al., 2008*). In this study we focused on drugs that: (1) have an inhibitory effect; (2) the majority of their targets are metabolic; (3) are not associated with dead-end reactions. The drug response data used in this analysis was measured in various ways: (a) ATP concentrations (HapMap dataset): In this case the *in silico* drug response is computed via MOMA in two steps; (1) obtaining a wild-type flux distribution via Flux Balance Analysis in which the corresponding drug target reaction is initially forced to be active (the pre-drug condition). Enforcing the target reaction to be active is necessary in order to get an effect on the resulting flux distribution following the inhibition simulated in the next step. Here we enforced a positive flux through the target reactions that is 50% of the maximal flux rate it is able to carry (our results are robust to various activation thresholds; *Supplementary file 1D*). (2) Next, the knock out flux distribution is computed via MOMA (*Segre et al., 2002*) while constraining the flux through the corresponding reactions to zero. This process is repeated for each personalized model separately and the predicted ATP production is used to estimate the cell response to the simulated drug. A robustness analysis is carried out by using 1000 different wild-type flux distributions (*Supplementary file 1D*); (b) AC50 values (CEU dataset): AC50 values represent the concentration in which the drug exhibits 50% of its maximum efficacy. In this case, *in silico* AC50 values are calculated by estimating the maximal flux rate carried by the target reaction when the growth rate is set to 50% of the drug's maximal response (a value that is available in the dataset used [*Lock et al., 2012*]); (c) IC50 values (NCI-60 dataset): IC50 values represent the concentration of drug needed in order to reduce the growth rate to 50% of its maximal value. In this case, *in silico* IC50 values are calculated by estimating the maximal flux rate carried by the target reaction when growth rate is set to 50% of its maximal value. In all cases of drug response

simulations the permutation test is carried out by permuting the measured data 1000 times and re-estimating the resulting correlation for each permuted vector.

## Predicting uptake and secretion rates

We have utilized the CORE data published by *Jain et al. (2012)* and normalized to cell size by *Dolfi et al. (2013)*, and compared it to uptake and secretion rates as predicted by the NCI-60 models. We have focused on 33 metabolites for which a corresponding exchange reaction exist in the human model and for which a non-zero flux was measured in at least three of the cell-lines. For each of these metabolites we estimated the maximal flux rate through its exchange reaction under at least 90% maximal growth rate, and compared it to that measured experimentally across the 59 cell-lines for which data exist. A similar approach was taken for both the PRIME and the E-Flux models. The permutation test is performed by permuting normalized CORE data 1000 times and repeating the process described above.

## Predicting differential effects on cancer cell growth

The effect of a reaction's deletion on cell growth in four breast cancer cell line models (MDA-MB-231, Hs578 t, BT549 and MDA-MB-435) was simulated via MOMA while enforcing the tested reaction to carry 50% of its maximal flux in the WT state (as described in the section 'Drug response simulations' above). The knock down of each tested reaction was simulated by inhibiting the target reaction by at least 75% of its maximal flux, then maximizing cellular growth under this perturbation. To increase specificity, we focused on reactions that are: (1) catalyzed by a single gene, and (2), their catalyzing gene does not catalyze more than three different reactions. Reactions were then ranked based on the variance in their knock down predicted growth rate across the four cell line models. 13 top predicted genes were selected for further experimental validation based on their high ranking in the list (i.e., high variance) and their association with diverse metabolic pathways (excluding transport reactions which their catalyzing enzymes are less specific). Each gene was examined experimentally in the two cell lines predicted to have the lowest and highest affect on cell growth. The permutation test is performed by permuting the models' predicted growth rates (after reaction knock down) 1000 times.

## Drug selectivity analysis

The effect of reaction's deletion on cell proliferation for the identification of selective treatment was simulated via MOMA with its robustness analysis as described in the section 'Predicting differential effects on cancer cell growth' above. The overlap between the set of cytostatic drug targets and the predicted non-selective targets was found to be robust to different thresholds that determine the value (in percentage) under which the deletion is considered to effect the cell's proliferation rate (*Supplementary file 1G*). The set of selective reaction targets is composed of those that reduce the growth of all normal cells by less than 20% and the growth of all cancer cells by more than 30%. Additionally, this set includes only those reactions that exhibit more than 20% difference in growth reduction between the normal and cancer proliferating cells (*Supplementary file 2*). Denoting growth inhibition as $Gi$ and growth survival as $Gs$, where $Gs$ is defined as $(1-Gi)$, the selectivity score computed for representation in *Figure 3B* is defined as $(Gi_{NCI60}-Gi_{HapMap})*Gs_{HapMap}$. The association between selective and non-selective targets and clinical survival data is performed by a Cox multivariate regression analysis. Specifically, a p-value for a Cox regression analysis of the expression of each gene and additional prognostic variables including patients' clinical stage, histological grade, tumor size, lymph node status and estrogen receptor status is computed. Each metabolic reaction is then assigned the minimal p-value achieved by its catalyzing enzymes. p-values are adjusted by Benjamini-Hochberg with FDR and $\alpha = 0.1$.

## Flux analysis for *MLYCD* knock down

Utilizing the RPMI-8226 model we first sampled the solution space and obtained 1000 wild-type flux distributions under maximal growth rate, in which the *MLYCD* reaction is forced to be active in a rate that is at least 50% of the maximal flux rate it can carry. Next, the knock down flux distribution is computed via MOMA while constraining the flux through the *MLYCD* reaction as described in 'Drug selectivity analysis' above. Utilizing the 1000 pre- and post-knockout flux distributions we applied a one-sided Wilcoxon ranksum test to determine reactions whose flux has been significantly increased/decreased. *Supplementary file 1H* summarizes these results.

### Reconstructing personalized models for clinical samples

The set of growth-associated reactions identified in the NCI-60 dataset was utilized as input to PRIME in the reconstruction process of the breast and lung cancer patients' models. PRIME then proceeds by adjusting the bounds of this set of reactions according to the specific cell expression levels.

## Experimental procedures

### Cell culture

HapMap cells (GM06997, CEPH/UTAH pedigree 13291) were obtained from Coriell Institute and RPMI-8226, K562, TK-10 and CAKI-1 cells were obtained from NCI-Frederick Cancer DCTD Tumor/Cell line Repository. HK2, MDA-MB-435, BT549, MDA-MB-231 and Hs578t cells were obtained from ATCC Repository. Cells were grown in RPMI 1640 plus 10% FBS in the presence of 5% carbon dioxide. Cell count was performed using CASY Cell Counter (Roche Applied Science). When indicated cells were incubated with 2 mM N-acetyl-cysteine. The breast cancer cell lines were cultured in RPMI (GIBCO, Life Technologies, Carlsbad, CA, USA) supplemented with 10% FBS (PAA, Pashing Austria) and 100 International Units/ml penicillin and 100 µg/ml streptomycin (Invitrogen, Carlsbad, CA, USA).

### Proliferation assay upon transient gene silencing

Cells were transfected and plated onto micro-clear 96-well plates (Greiner Bio-one, Monroe, NC, USA). Human mix of four singles siRNAs (SmartPool) for the 13 predicted genes were purchased in siGENOME format from Dharmacon (Lafayette, CO, USA). A custom-made siRNA targeting luciferase (siLUC) was used as negative control and also purchased from Dharmacon (Lafayette, CO, USA). Plates were diluted to 1 µM working concentration in complementary 1× siRNA buffer in a 96-well plate format. A 50 nM reverse transfection was performed according to manufacturer's guidelines using INTERFERin as transfection reagent. Complex time was 20 min and 10,000, 6000, 7000 and 6000 of respectively MDA-MB-435, BT549, MDA-MB-231 and Hs578t cells were added. The plate was placed in the incubator overnight and the medium was refreshed the following morning. After a total of 5 days of incubation, the cells were stained live with Hoechst (nr. 33342) and fixed with TCA (Trichloroacetic acid). Whole wells were imaged using epi-fluorescence and the number of nuclei was determined using a custom-made ImagePro macro. The results were expressed as percentage of growth inhibition when compared to the negative control siLUC. This proliferation assay was performed in triplicate (one well per gene knock down, per cell line and per replicate).

### Silencing of *MLYCD*

#### siRNA

$2 \times 10^6$ cells were nucleofected using Nucleofector I (Amaxa) and Amaxa Cell Line Nucleofector Solution Kit C (Lonza), program A-030 and 1 µM siRNA. The MLYCD-targeting siRNA constructs were purchased from Sigma Aldrich and are as follows: siRNA1: GUACCUACAUCUUCAGGAA; siRNA2: CAAAGUUGACUGUGUUCUU; siRNA3: GAAGGAACAUCCUCCAUCA. The non-targeting siRNA is the MISSION siRNA Universal Negative Control #1 (Sigma Aldrich).

#### shRNA

The viral supernatant for infection was obtained from the filtered growth media of the packaging cells HEK293T transfected with with 3 µg psPAX, 1 µg pVSVG, 4 µg of shRNA contructs and 24 µl Lipofectamine 2000 (Life Technology) and the relevant shRNAs. $5 \times 10^5$ cells were then plated on 6-well plates and infected with the viral supernatant in the presence of 4 µg/ml polybrene. After 2 days, the medium was replaced with selection medium containing 2 µg/ml puromycin.

The expression of the shRNA constructs was induced by incubating cells with 2 µg/ml Doxycyclin.

The shRNA sequence were purchased from Thermoscientific and are as follows: shRNA1: TTCTG AAGCACTTCACACG; shRNA2: GATTTTGTTCTTCTCTTCT; shRNA NTC #RHS4743.

### Glutathione measurements

Glutathione levels were measured using GSH-Glo Glutathione Assay (Promega) after 72 hr of Doxyclyclin induction, using 20 µl/well of $2 \times 10^5$ cells/ml, following to the manufacturer's instructions.

### qPCR experiments

mRNA was extracted with RNeasy Kit (Qiagen) and 1 µg of mRNA was retrotranscribed into cDNA using High Capacity RNA-to-cDNA Kit (Applied Biosystems, Life Technologies, Paisley, UK).

For the qPCR reactions 0.5 µM primers were used. 1 µl of Fast Sybr green gene expression master mix; 1 µl of each primers and 4 µl of 1:10 dilution of cDNA in a final volume of 20 µl were used. Real-time PCR was performed in the Step One Real-Time PCR System (Life Technologies Corporation Carlsbad, California) using the fast Sybr green program and expression levels of the indicated genes were calculated using the ΔΔCt method by the appropriate function of the software using actin as calibrator.

Primer sequences are as follows:

**MLYCD**: Fwd: ttgcacgtggcactgact; RV: ggatgttccttcacgattgc; **Actin**: QuantiTect primer QT00095431 (Qiagen), sequence not disclosed.

## Isotope tracing experiments

$2 \times 10^5$ cells/ml cells were seeded in six well plates. After 48 hr cells were rapidly pelleted and media was replaced with labeled nutrients-containing media. For 1,2-$^{13}$C-Glucose and $^{13}C_6$-Glucose experiments labeled compounds were dissolved in glucose-free RPMI 1640 medium supplemented with 10% Fetal Bovine Serum media to a final concentration of labeled glucose of 11 mM. $^{13}$C-Palmitate was dissolved in EtOH to a final concentration of 20 mM, mixed with a 10% Bovine Serum Albumin solution at a 1:5 ratio and incubated 1 hr at 37°C. After incubation the $^{13}$C-Palmitate solution was diluted in serum-containing RPMI 1640 medium to a final concentration of 50 µM. The cells were incubated with labeled nutrients-containing media for 24 hr after which metabolites were extracted and analyzed with LC-MS as described below. All labelled metabolites were purchased at CKGas Products Ltd (UK).

## Metabolomic extraction of cell lines

$5 \times 10^5$ cells/ml were plated onto six-well plates and cultured in standard medium for 24 hr. For the intracellular metabolomic analysis, cells were quickly washed for three times with phosphate buffer saline solution (PBS) to remove contaminations from the media. The PBS was thoroughly aspirated and cells were lysed by adding a pre-cooled Extraction Solution (Methanol:Acetonitrile:Water 50:30:20). The cell number was counted and cells were lysed in 1 ml of ES per $2 \times 10^6$ cells. The cell lysates were vortexed for 5 min at 4°C and immediately centrifuged at 16,000×$g$ for 15 min at 0°C.

## LC-MS metabolomic analysis

For the LC separation, column A was the Sequant Zic-Hilic (150 mm × 4.6 mm, internal diameter (i.d.) 5 µm) with a guard column (20 mm × 2.1 mm i.d. 5 µm) from HiChrom, Reading, UK. Mobile phase A: 0.1% formic acid vol/vol in water. Mobile B: 0.1% formic acid vol/vol in acetonitrile. The flow rate was kept at 300 µl/min and gradient was as follows: 0 min 80% of B, 12 min 50% of B, 26 min 50% of B, 28 min 20% of B, 36 min 20% of B, 37–45 min 80% of B. Column B was the sequant Zic-pHilic (150 mm × 2.1 mm i.d. 5 µm) with the guard column (20 mm × 2.1 mm i.d. 5 µm) from HiChrom, Reading, UK. Mobile phase C: 20 mM ammonium carbonate plus 0.1% ammonia hydroxide in water. Mobile phase D: acetonitrile. The flow rate was kept at 100 µl/min and gradient as follow: 0 min 80% of D, 30 min 20% of D, 31 min 80% of D, 45 min 80% of D. The mass spectrometer (Thermo Q-Exactive Orbitrap) was operated in a polarity switching mode.

## Datasets

Expression data and growth rate measurements for the HapMap dataset were taken from (*Choy et al., 2008*). The data includes Utah residents with Northern and Western European ancestry (CEU; 56 samples), Han Chinese in Beijing, China (CHB; 43 samples), Japanese in Tokyo, Japan (JPT; 43 samples) and Yoruba from Ibadan, Nigeria (YRI; 82 samples). Expression data for the NCI-60 dataset was taken from (*Lee et al., 2007*). Doubling times for the NCI-60 cell lines were downloaded from the website of the Developmental Therapeutics Program (DTP) at NCI/NIH (http://dtp.nci.nih.gov/docs/misc/common_files/cell_list.html).

## Acknowledgements

The authors would like to thank Yoav Teboulle, Noa Cohen, Allon Wagner, Matthew Oberhardt, Adam Weinstock, Roded Sharan, Tamar Geiger and Ayelet Erez for their comments on the manuscript and helpful discussions. KY is partially supported by a fellowship from the Edmond J. Safra Bioinformatics center at Tel-Aviv University and is grateful to the Azrieli Foundation for the award of an Azrieli Fellowship. YYW is partially supported by Eshkol Fellowship (Israeli Ministry of Science and Technology). This research is supported by a grant from the Israeli Science Foundation (ISF) and Israeli Cancer

Research Fund (ICRF) to ER and by the I-CORE Program of the Planning and Budgeting Committee and The Israel Science Foundation (grant No 41/11).

## Additional information

### Funding

| Funder | Grant reference number | Author |
| --- | --- | --- |
| Israel Science Foundation | 0603804571 | Keren Yizhak, Eytan Ruppin |
| Israel Cancer Research Fund | 0603804521 | Keren Yizhak, Eytan Ruppin |
| Israel Science Foundation | I-CORE Program Grant No 41/11 | Keren Yizhak, Eytan Ruppin |
| Medical Research Council | | Edoardo Gaude, Christian Frezza |

The funders had no role in study design, data collection and interpretation, or the decision to submit the work for publication.

### Author contributions

KY, Conception and design, Acquisition of data, Analysis and interpretation of data, Drafting or revising the article; EG, Acquisition of data, Analysis and interpretation of data, Drafting or revising the article; SLD, Acquisition of data, Analysis and interpretation of data; YYW, Conception and design, Acquisition of data, Drafting or revising the article; GYS, BW, CF, Analysis and interpretation of data, Drafting or revising the article; ER, Conception and design, Drafting or revising the article

## Additional files

### Supplementary files

• Supplementary file 1. Supplementary files 1A–K.

• Supplementary file 2. Supplementary files 2A–D.

• Supplementary file 3. The PRIME code.

• Supplementary file 4. The HapMap and NCI-60 models as SBML.

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
