## [Decision Letter]

Thank you for sending your work entitled “Phenotype-based cell-specific metabolic modeling reveals metabolic liabilities of cancer” for consideration at *eLife.* Your article has been evaluated by Charles Sawyers (Senior editor) and 4 reviewers, one of whom, Chi Dang, is a member of our Board of Reviewing Editors. Our opinion is favorable but there are a number of major concerns that you will need to address before we can consider the paper for acceptance at *eLife*.

The Reviewing editor and the other reviewers discussed their comments before we reached this decision, and the Reviewing editor has assembled the following comments to help you prepare a revised submission.

The authors present a novel method termed PRIME for building cell-specific Genome Scale Models and apply this technique for the reconstruction of metabolic models of normal and cancer cells. The work is interesting as it addresses development of technique that is able to predict metabolic phenotype, a problem currently lacking conclusive answers. However, because of its highly specialized nature, whether the paper would be useful to cancer biologists is unclear. Even if the authors computationally predicted 1 target that slows the growth of some cancer cell lines, would this represent a significant advance? How many targets would need to be validated to provide a false discovery rate or the true utility of the approach? In the absence of convincing biologic data, the computational methods become less relevant.

In this regard, additional experimental data should be provided. Specifically, the authors use siRNA to knockdown of MLYCD without providing any assays to determine if the knockdown was effective other than showing a modest suppression in total mRNA content that results in a modest suppression in cell growth. The authors need to assay the cells MLYCD activity, de novo fatty acid synthesis and fatty acid oxidation. The authors instead use steady-state metabolomics, which can be very misleading. For example, they go on to suggest that the mechanism of the selective toxicity results from a redox imbalance for which the argument (and data) are extremely weak. Furthermore, in humans, mutations in MLYCD cause malonic and methylmalonic aciduria (OMIM#248360) therefore the selective toxicity may be a cell culture artifact.

Another major concern is the lack of accompanying models or at least the base code of the method, which would facilitate easier dissemination. The authors can submit their code as supplementary documentation. It would also be great, if at all possible, to make the cell-specific GSMMs available for download, either from the journal's website, from a group resource, or from public repositories.

The Methods section entitled “The PRIME algorithm”, while overall well-written and presenting the basics of PRIME effectively, seems to have mixed up notation, because the text alternates between using i,j and t,p for reaction and sample sets, respectively. This can be particularly confusing since the i,j combination was used for the metabolite, reaction sets just in the previous section. Perhaps the authors were trying to fix that exact discrepancy, but the symbols were only replaced in part of the text. This should be made consistent throughout.

Not an absolute requirement, but rather a suggestion: the PRIME method presented here makes heavy use of MOMA. Since the authors take the time and space to briefly present and explain the FBA formulation, perhaps a similar short section on MOMA would be useful for the reader.

[Editors' note: further revisions were requested prior to acceptance, as described below.]

Thank you for resubmitting your work entitled “Phenotype-based cell-specific metabolic modeling reveals metabolic liabilities of cancer” for further consideration at *eLife*. Your revised article has been favorably evaluated by Charles Sawyers (Senior editor), a Reviewing editor, and the other three original reviewers. The manuscript has been improved but there are some remaining issues that need to be addressed before acceptance, as outlined below:

1) It's improper when one presents only one or a few metabolite isotopologues (mass-isotopomers). Only whole metabolite mass-isotopomer distribution (MID) ranging from m+0 to m+n (where n- number of C atoms in the molecule) makes sense. Please completely describe your observed MID for TCA cycle intermediates and palmitate isotopologues (Figure 5 and Figure 5—figure supplement 5) or at least describe that other mass-isotopomers were not observable.

2) It is not clear why m+1/m+2 isotopologue ratios for pyruvate and lactate are different; assuming that lactate dehydrogenase usually operates at near equilibrium condition. Please clarify.

---

## [Author Response]

Addressing the main comments raised by the reviewers, we have now:

1) Comprehensively investigated the functional effects of the silencing of MLYCD using ^13^C labeling experiments.

2) Validated the NCI-60 PRIME models by additional datasets of uptake and secretion rates.

3) Provided the full PRIME code and the generated HapMap and NCI-60 models.

*The authors present a novel method termed PRIME for building cell-specific Genome Scale Models and apply this technique for the reconstruction of metabolic models of normal and cancer cells. The work is interesting as it addresses development of technique that is able to predict metabolic phenotype, a problem currently lacking conclusive answers. However, because of its highly specialized nature, whether the paper would be useful to cancer biologists is unclear. Even if the authors computationally predicted 1 target that slows the growth of some cancer cell lines, would this represent a significant advance? How many targets would need to be validated to provide a false discovery rate or the true utility of the approach? In the absence of convincing biologic data, the computational methods become less relevant*.

We thank the reviewers for this remark. Indeed, in this study we worked on developing a computational approach that will enable the generation of cell-specific models with a significant predictive value. The main thrust of the paper is therefore focused on the validation of the reconstructed models through different publically available resources, including growth rates, drug response, biomarkers and clinical survival data, and through the generation of new data concerning the differential effect of 13 predicted genes upon cancer cell growth. The significant results obtained by these various validations vouch for its ability to provide meaningful predictions for yet untested targets. The specific case of *MLYCD* as a selective drug target is tested here experimentally more deeply, and further data has been added to its validation as described in our response to the next comment.

*In this regard, additional experimental data should be provided. Specifically, the authors use siRNA to knockdown of MLYCD without providing any assays to determine if the knockdown was effective other than showing a modest suppression in total mRNA content that results in a modest suppression in cell growth. The authors need to assay the cells MLYCD activity, de novo fatty acid synthesis and fatty acid oxidation. The authors instead use steady-state metabolomics, which can be very misleading*.

We agree with the referees that steady-state metabolomics measurements can be misleading if used to infer the activity of metabolic pathways. Therefore, to address the referee’s concerns we have performed ^13^C-labeling experiments followed by targeted LCMS analyses to assess fatty acid synthesis and fatty acid oxidation in MLYCD-depleted cells. Although we were not able to reliably detect malonyl-CoA and malonate in our experiments, we have now shown that the silencing of MLYCD leads to a significant decrease in fatty acid oxidation (Figure 5) and to increased *de novo* fatty acid synthesis (Figure 5). These results demonstrate that the extent of depletion of MLYCD achieved.

*For example, they go on to suggest that the mechanism of the selective toxicity results from a redox imbalance for which the argument (and data) are extremely weak*.

We thank the referees for this remark, which prompted us to better support our hypothesis. It is worth mentioning that we investigated the deregulation of redox homeostasis as an experimental validation of PRIME’s predictions on the global metabolic effects of MLYCD depletion. These predictions indicated that the silencing of MLYCD causes a redox imbalance due to an increased demand of NAPDH to support accelerated lipid synthesis. We have now strengthened this hypothesis by: (1) confirming that fatty acid synthesis is increased in MLYCD-depleted cells (see comment above); and (2) measuring the activity of the pentose phosphate pathway, the major source of cytosolic NADPH [1]. To this aim, we performed an isotope tracing experiment with 1,2-^13^C_2_-Glucose, which enables us to distinguish between glycolysis and PPP fluxes (Figure 5). Reassuringly, we observed a significant increase in the PPP upon MLYCD inhibition (Figure 5). These results are in line with the hypothesis that aberrant fatty acid synthesis drains redox equivalents, exposing cells to higher oxidative stress.

*Furthermore, in humans, mutations in MLYCD cause malonic and methylmalonic aciduria (OMIM#248360) therefore the selective toxicity may be a cell culture artifact*.

The referees are correct in pointing out that, in humans, the loss of MLYCD leads to methylmalonic aciduria, an extremely rare autosomal recessive disorder. Nevertheless, *in vivo* experiments in rodents and pigs [2, 3], *ex vivo* experiments in human skeletal muscle [4], and in MRC-5 non-transformed fibroblasts [5] suggest that the inhibition of MLYCD is well tolerated, as our results indicate. It is therefore possible that the inhibition of the enzyme has no detrimental effects on normal cells and tissues, and that other factors contribute to the severity of MLYCD deficiency in humans, including a toxic effect of the mutated protein [6]. We agree with the referees that more experiments are required to assess the therapeutic window of MLYCD inhibition *in vivo*, but we believe that these experiments are beyond the scope of this manuscript. In light of the referees’ comment, however, a brief discussion along these lines has now been added.

*Another major concern is the lack of accompanying models or at least the base code of the method, which would facilitate easier dissemination. The authors can submit their code as supplementary documentation. It would also be great, if at all possible, to make the cell-specific GSMMs available for download, either from the journal's website, from a group resource, or from public repositories*.

The PRIME code together with a detailed README file is provided now as [Supplementary-material SD3-data]. Furthermore, we also provide the entire set of HapMap and NCI-60 model as [Supplementary-material SD4-data]. A reference to these files appears in the main text.

*The Methods section entitled “The PRIME algorithm”, while overall well-written and presenting the basics of PRIME effectively, seems to have mixed up notation, because the text alternates between using i,j and t,p for reaction and sample sets, respectively. This can be particularly confusing since the i,j combination was used for the metabolite, reaction sets just in the previous section. Perhaps the authors were trying to fix that exact discrepancy, but the symbols were only replaced in part of the text. This should be made consistent throughout*.

*Not an absolute requirement, but rather a suggestion: the PRIME method presented here makes heavy use of MOMA. Since the authors take the time and space to briefly present and explain the FBA formulation, perhaps a similar short section on MOMA would be useful for the reader*.

We thank the reviewers for drawing our attention to that issue. We have now fixed the text accordingly as follows: *i* and *j* represent row and column indices in the stoichiometric matrix, respectively; *t* and *p* represent the number of rows and columns in the Exp-matrix, respectively; *a* and *b* represent row and column indices in the Exp-matrix, respectively. Additionally, following the reviewer's suggestion, a detailed description of MOMA was added to the Methods as follows: “Gene knock outs are simulated by constraining the flux through the corresponding metabolic reaction to zero. Following, two different approaches can be taken to estimate the effect of a perturbation on the network: (1) via Flux Balance Analysis (FBA) [7] where maximization of growth rate is defined as the cellular objective function (maxVbio); (2) Minimization of Metabolic Adjustment (MOMA) [8] where the minimization of the Euclidean distance between a wild-type flux distribution (Vwt) and the post-perturbation flux distribution (VKO) is set as the cellular objective function min∑in(Vwt,i−VKO,i)2. Different wild-type flux distributions are obtained via sampling where each sample is determined based on a FBA analysis maximizing for cellular growth.”

*[Editors' note: further revisions were requested prior to acceptance, as described below*.*]*

*1) It's improper when one presents only one or a few metabolite isotopologues (mass-isotopomers). Only whole metabolite mass-isotopomer distribution (MID) ranging from m+0 to m+n (where n- number of C atoms in the molecule) makes sense. Please completely describe your observed MID for TCA cycle intermediates and palmitate isotopologues (*Figure 5
*and*
Figure 5—figure supplement 5*) or at least describe that other mass-isotopomers were not observable*.

This point is well taken. We agree with the referees that all the mass isotopologues of a given metabolites should be indicated in the manuscript. In the presented experimental conditions, we were not able to detect the isotopologues of TCA cycle intermediates above m+2 in the ^13^C-palmitate experiment (Figure 5) and of palmitate above m+10 in the ^13^C-glucose labeling experiment (Figure 5). Nevertheless, for completeness, we have now added the following analyses:

A) The m+0 to m+2 isotopologues of the TCA cycle intermediates upon incubation with ^13^C_16_-palmitate (new Figure 5—figure supplement 3)

B) The m+0 to m+10 isotopologues of palmitate from the ^13^C-glucose labeling experiment (updated Figure 5)

C) The m+0 to m+6 isotopologues of the TCA cycle intermediates from the ^13^C-glucose labeling experiment (updated Figure 5—figure supplement 5, now Figure 5—figure supplement 6).

We have also described in more details the results of the experiments in the light of the new data presentation and clarified in the figure legend that certain isotopologues were not detectable in our analyses.

*2) It is not clear why m+1/m+2 isotopologue ratios for pyruvate and lactate are different; assuming that lactate dehydrogenase usually operates at near equilibrium condition. Please clarify*.

We agree with the referees that these results may generate some confusion. The discrepancy between the labeling of pyruvate and lactate may be due to different pools of pyruvate being diverted to lactate and alanine, as previously observed (Zwingmann C et al. Glia. 2001 May;34(3):200-12). However, we believe that this discussion is beyond the scope of the manuscript and, to avoid confusion, we decided to remove the pyruvate labeling, and to present only the m+1/m+2 ratio for lactate, as it is generally accepted for understanding the flux of carbons through the PPP.

References

1. Fan, J., et al., *Quantitative flux analysis reveals folate-dependent NADPH production.* Nature, 2014. **advance online publication**.

2. Dyck, J.R.B., et al., *Malonyl Coenzyme A Decarboxylase Inhibition Protects the Ischemic Heart by Inhibiting Fatty Acid Oxidation and Stimulating Glucose Oxidation.* Circulation Research, 2004. **94**(9): p. e78-e84.

3. Wu, H., et al., *Effect of Inhibiting Malonyl-CoA Decarboxylase on Cardiac Remodeling after Myocardial Infarction in Rats.* Cardiology, 2014. **127**(4): p. 236-244.

4. Bouzakri, K., et al., *Malonyl CoenzymeA Decarboxylase Regulates Lipid and Glucose Metabolism in Human Skeletal Muscle.* Diabetes, 2008. **57**(6): p. 1508-1516.

5. Zhou, W., et al., *Malonyl-CoA decarboxylase inhibition is selectively cytotoxic to human breast cancer cells.* Oncogene, 2009. **28**(33): p. 2979-2987.

6. Polinati, P.P., L. Valanne, and T. Tyni, *Malonyl-CoA decarboxylase deficiency: Long-term follow-up of a patient new clinical features and novel mutations.* Brain and Development, (0).

7. Varma , A. and B.O. Palsson, *Metabolic flux balancing: Basic concepts, scientific and practical use.* Nature Biotechnology, 1994. **12**: p. 994-998.

8. Segre, D., D. Vitkup, and G.M. Church, *Analysis of optimality in natural and perturbed metabolic networks.* Proceedings of the National Academy of Sciences, 2002. **99**: p. 15112 - 15117.